# The validity and reliability of the Dutch version of the Student Satisfaction and Self-Confidence in Learning Scale (SCLC) for pharmacy technicians

**Narin Akrawi**◉\*, **Wim J.R. Rietdijk, Birgit C.P. Koch, Floor van Rosse, Heleen van der Sijs**

Department of Hospital Pharmacy, Erasmus MC University Medical Center, Rotterdam, The Netherlands

\* n.martens-akrawi@erasmusmc.nl

## Abstract

### Background

The Student Satisfaction and Self-Confidence in Learning Scale (SCLC) is widely used to measure satisfaction and self-confidence in learning. The 13-item scale includes two subscales: satisfaction with education (5 items) and self-confidence in learning (8 items), rated on a 5-point Likert scale. However, no validated Dutch version existed for pharmacy technicians, a group increasingly involved in complex healthcare roles. This study aimed to translate, adapt, and validate the SCLC for use among Dutch pharmacy technicians.

### Methods

The SCLC was translated into Dutch following cross-cultural adaptation guidelines, including forward and back-translation by three bilingual experts. The questionnaire was administered to pharmacy technicians at Erasmus MC. Internal consistency was assessed using Composite Reliability (CR) and Average Variance Extracted (AVE). A confirmatory factor analysis (CFA) evaluated construct validity.

### Results

A total of 129 pharmacy technicians completed the questionnaire. CFA indicated a good fit for a two-factor model, with a statistically significant Chi-square (p < 0.000), and a Chi-square/df ratio of 2.5. Fit indices, including CFI (0.92) and RMSEA (0.102), suggested moderate model fit.

### Conclusion

This study suggests that the Dutch version of the SCLC is a moderately reliable and valid tool for assessing pharmacy technicians' satisfaction with education and self-confidence in learning. Given the absence of other validated instruments in

**Data availability statement:** All relevant data are within the paper and its Supporting Information files.

**Funding:** The author(s) received no specific funding for this work.

**Competing interests:** The authors have declared that no competing interests exist.

this context, this scale offers a useful starting point for evaluating and improving educational programs. However, as the psychometric properties indicated room for improvement and the model fit was moderate, further research with larger samples is needed to refine and confirm the suitability of this questionnaire.

## Introduction

In the ever-evolving landscape of healthcare, continuous professional development is paramount to ensure that healthcare professionals, including pharmacy technicians, remain up to date on the latest developments and best practices [1–3]. Lifelong learning plays a pivotal role in this endeavor, enabling pharmacy technicians to enhance their knowledge and skills through various educational activities [4–6]. Therefore, we recognize the importance of ongoing education and developed digital learning modules tailored to the specific needs of pharmacy technicians working in the hospital, facilitating their professional growth and efficacy in delivering patient-centered care. In addition to these newly developed digital modules, pharmacy technicians also participate in clinical lessons, which have been an existing component of their training. By aligning our digital learning modules with the daily practice in which pharmacy technicians work, we aim to equip them with the fundamental skills necessary for thriving in the dynamic healthcare environment.

One key aspect is satisfaction with education, which is crucial for healthcare professionals. It enables them to stay current in their field, advance in their careers, and maintain and enhance their competency in delivering high-quality patient care. Moreover, it fosters their motivation and engagement, encouraging lifelong learning and skill improvement [7]. Well-educated and satisfied healthcare professionals are also known to be better equipped to handle complex medical situations, reducing the likelihood of medical errors and enhancing patient safety [8]. Lastly, healthcare professionals who are satisfied with their education are more likely to experience overall job satisfaction and are less likely to leave their positions [9]. This contributes to a stable workforce and continuity of care for patients.

Another crucial aspect for healthcare professionals is self-confidence, which denotes the assurance and trust in one's capability, while self-efficacy reflects beliefs in one's ability to perform specific tasks or behaviors [10,11]. In educational contexts, both constructs are vital, as they influence how learners perceive their ability to acquire and apply knowledge in clinical practice. In this regard, the Student Satisfaction and Self-Confidence in Learning Scale (SCLC), developed by the National League for Nursing (NLN), is a widely used instrument for evaluating satisfaction with learning activities and perceived self-confidence in learning outcomes [10–12].

The SCLC specifically captures two interrelated dimensions: (1) student satisfaction, which refers to learners' contentment with the structure, relevance, and quality of the learning activity; and (2) self-confidence in learning, defined as the learner's belief in their capacity to understand and apply what they have learned [11,13]. These dimensions are recognized as essential indicators of educational effectiveness in

healthcare training, as they reflect not only the learner's perception of the learning experience but also their readiness to transfer acquired knowledge to practice [14–17].

Validation studies across several countries—including Spanish, Portuguese, Turkish, Norwegian, Polish and Chinese translations—have demonstrated the psychometric robustness of the scale and its relevance across cultural and professional contexts [14–19]. For example, the Spanish validation [17] emphasized that the SCLC reflects the motivational and cognitive components of learning, and is strongly associated with enhanced learner engagement, deeper learning, and greater integration of clinical competencies. Furthermore, increased satisfaction and confidence in learning have been linked to improved academic performance, reduced anxiety, and stronger commitment to lifelong learning [7,13,14,16].

By assessing both satisfaction and self-confidence in the context of professional education, the SCLC provides valuable insight into learners' perceived preparedness and motivation. This, in turn, has implications for their performance in practice settings and overall job satisfaction [20,21]. Thus, measuring these constructs supports targeted improvements in healthcare education and contributes to professional growth, workforce stability, and improved patient care outcomes [8,9,22].

Although the SCLC has been widely used and validated in nursing education, no validated instrument currently exists that is specifically suited for pharmacy technicians and translated into Dutch. Therefore, we aim to utilize the SCLC to assess the effects of our digital learning modules on pharmacy technicians' self-confidence and satisfaction with learning. By doing so, we aim to fill this gap and provide a reliable tool for evaluating educational experiences within this specific professional group.

While the SCLC was originally developed to assess learning in standardized patient simulations, its structure allows for adaptation to a broader range of educational contexts [11,13,17].

Several studies have shown that, with minor adjustments to simulation-specific wording, the instrument remains valid and relevant for evaluating learners' satisfaction and self-confidence across diverse learning environments. In our study, we chose to retain the original simulation-related phrasing, as these items continue to offer valuable insights into pharmacy technicians' engagement with practical, real-world learning experiences. Given that both nurses and pharmacy technicians complete vocational education and receive workplace-based training, the SCLC is considered appropriate for use within this group. Translating, adapting, and validating the SCLC for Dutch pharmacy technicians thus offers a valuable opportunity to address the current lack of context-specific tools and to support the evaluation of educational interventions within this profession.

The aim of this study is to adapt, translate and validate the SCLC for Dutch pharmacy technicians. By doing so, we strive to offer a reliable tool for assessing satisfaction with learning activities and self-confidence in this professional group. These insights will help to support continuous professional development, enhance job satisfaction, and ultimately contribute to the delivery of high-quality, patient-centered care.

## Methods

### Study setting

The study was conducted at Erasmus Medical Center, which is a large academic medical center with 1233 beds and over 16,000 employees. In 2021, the hospital conducted nearly 800,000 outpatient consultations. As a tertiary care center, Erasmus MC specializes in complex and highly specialized healthcare services, providing advanced treatment for rare diseases and critical conditions.

The role of the pharmacy technician at Erasmus MC is crucial in supporting medication management and pharmaceutical care. Pharmacy technicians assist in preparing, dispensing, and managing medications, ensuring proper drug dosages and monitoring for potential drug interactions. They collaborate with pharmacists and other healthcare professionals to deliver optimal care to patients, particularly in complex, tertiary healthcare settings. Their responsibilities also include patient education regarding the safe use of medications and maintaining accurate medication records. At the time of this

study, 230 pharmacy technicians were employed across both the clinical pharmacy and outpatient pharmacy departments of Erasmus MC. Pharmacy technicians working in either of these departments during the study period were recruited for this study.

## Recruitment

Recruitment was conducted via email and verbally during work meetings. The recruitment process included both newly hired technicians and those with longer-term employments at the clinical pharmacy and outpatient pharmacy departments of the hospital. We aimed for a sample size of over 200, following the recommendations of Kyriazos and Schreiber et al., who both suggest having at least 10 cases per indicator [23,24]. Recruitment took place between September 4, 2023, and February 29, 2024. Participants provided written digital informed consent prior to completing the online questionnaire.

## Translation and adaptation of the SCLC questionnaire

The SCLC consists of 13 items, categorized into two dimensions: ''satisfaction with instruction'' and "self-confidence in learning". The "satisfaction with instruction" subscale includes five items that evaluate satisfaction with teaching methods, diversity of learning materials, facilitation, motivation, and overall suitability of simulation. The "self-confidence with learning" subscale comprises eight items that gauge self-confidence in content mastery, content necessity, skills development, available resources, and knowledge of how to obtain help to solve clinical problems in simulation. Participants indicate their personal feelings about each statement using a Likert-style scale, ranging from 1) strongly disagree to 5) strongly agree.

The SCLC questionnaire was translated from English to Dutch, with adjustments made to better fit our context of evaluating various forms of education, including digital learning modules where no teacher is involved. The adjustments included changing 'teacher' to 'instructor' and 'simulation' to 'learning activities. In the version of the questionnaire adapted to the Dutch language and context, we included the following clarification: ''By 'instructor' we mean the person responsible for delivering or guiding the learning activity.'' Furthermore, we specified that the term 'learning activity' could refer to both a classroom-based lesson and digital modules.

Three bilingual individuals proficient in both English and Dutch independently translated the questionnaire, followed by a back-translation to ensure conceptual equivalence [25,26]. Any discrepancies or ambiguities encountered during the translation process were resolved through consensus among the translators, although no such issues arose in practice. The modified SCLC and its Dutch translation are presented in Table 1.

Lime Survey Version 6.8.2 was used to administer the survey digitally and participants had the opportunity to submit partially completed questionnaires.

## Ethical considerations

The Medical Ethics Committee of Erasmus MC determined that the Medical Research Involving Human Subjects Act (WMO) was not applicable to this study and granted a 'waiver' from the requirements normally imposed by the WMO (MEC-2023–0155).

The participants were briefed on the objectives of the study and were guaranteed confidentiality for all collected data. They contributed to this study voluntarily. We also assured the participants that their responses would not affect their daily work and that the answers were submitted anonymously, ensuring they could not be linked back to the individual.

The translation and adaptation of the instrument to non-simulation contexts have been conducted with the permission of the National League for Nursing (NLN); however, NLN does not assume responsibility for the accuracy of the translation or the modifications made for non-simulation use. Any inquiries regarding the Dutch version of the instrument should be directed to the NLN. For further details about research instruments and copyright, please visit the NLN website [27]. The NLN retains the copyright for both the original English version and the Dutch translation of the instrument.

**Table 1. The presentation of the adapted version of the SCLC questionnaire from English to Dutch.**

| English | Dutch | |
|---|---|---|
| **Satisfaction with current learning** | **Tevredenheid met het huidige leren** | Scores P1<br>items 1–5:<br>minimum score = 5, maximum score = 25 |
| ITEM 1. the learning activities were useful and effective | ITEM 1. De leeractiviteiten waren nuttig en effectief | |
| ITEM 2. the learning activities provided me with a variety of learning materials and activities to enhance my pharmaceutical knowledge. | ITEM 2. De leeractiviteiten boden me een verscheidenheid aan leermaterialen en activiteiten om mijn farmaceutische kennis te vergroten | |
| ITEM 3. I enjoyed the way the teaching was delivered. | ITEM 3. Ik vond de manier waarop het onderwijs werd gegeven leuk | |
| ITEM 4. the teaching methods were motivating and helped me to learn. | ITEM 4. De onderwijsmethoden waren motiverend en hebben mij geholpen om te leren | |
| ITEM 5. the way I received the education was suitable for my learning style. | ITEM 5. De manier waarop ik het onderwijs kreeg, was geschikt voor de manier waarop ik leer | |
| **Self-Confidence in learning** | **Zelfvertrouwen in het leren** | Scores P2<br>Items 6–13:<br>minimum score = 8, maximum score = 40 |
| ITEM 6. I am confident that I have mastered the content of the learning activities. | ITEM 6. Ik heb er vertrouwen in dat ik de inhoud van de leeractiviteiten goed onder de knie heb | |
| ITEM 7. I am confident that important topics necessary for performing well in my hospital work have been covered in the learning activities I have had. | ITEM 7. Ik heb er vertrouwen in dat in de leeractiviteiten die ik heb gehad, belangrijke onderwerpen zijn behandeld, die noodzakelijk zijn om mijn werk in het ziekenhuis goed uit te voeren | |
| ITEM 8. I am confident that I am developing the knowledge and acquiring the required skills from the learning activities for my duties as a pharmacy technician. | ITEM 8. Ik heb er vertrouwen in dat ik de kennis ontwikkel en de vereiste vaardigheden uit de leeractiviteiten verkrijg voor mijn taken als apothekersassistent | |
| ITEM 9. the instructors/teachers used useful resources to teach me things. | ITEM 9. De instructeurs/docenten gebruikten nuttige bronnen om mij dingen te leren | |
| ITEM 10. it is my responsibility as a pharmacy technician to learn what is required from the learning activities. | ITEM 10. Het is mijn verantwoordelijkheid als apothekersassistent om te leren wat nodig is van de leeractiviteiten. | |
| ITEM 11. I know where to seek help when I don't understand the material being offered. | ITEM 11. Ik weet waar ik hulp moet vragen, wanneer ik de aangeboden stof niet goed begrijp | |
| ITEM 12 I know how to use learning activities to solve problems. | ITEM 12. Ik weet hoe ik leeractiviteiten moet gebruiken om vraagstukken op te lossen | |
| ITEM 13 It is the responsibility of the instructor/teacher to tell me what I need to learn from the learning activities. | ITEM 13. Het is de verantwoordelijkheid van de instructeur/docent om mij te vertellen wat ik moet leren uit de leeractiviteiten. | |

This table shows the adapted English version of the SCLC questionnaire, in which terms such as "instructor" and "simulation" have been modified to fit non-simulation contexts, alongside the Dutch translation. Additionally, it presents the distribution of the items within each dimension: 'Satisfaction with current learning' (minimum score = 5, maximum score = 25) and 'Self-Confidence in learning' (minimum score = 8, maximum score = 40), along with the corresponding items in both the adapted English and Dutch versions.

## Assumption checks

Prior to performing Confirmatory Factor Analysis (CFA), key assumptions were assessed to ensure the appropriateness of the analysis. Normality was examined using histograms and normal probability plots for all items in SPSS. The data showed no substantial skewness or kurtosis, indicating an approximately normal distribution. In addition, multivariate

outliers were assessed using Cook's Distance, which did not identify any influential cases. While linearity and multicol-linearity were not formally tested, theoretical justification and inter-item correlations indicated no major concerns [28]. Based on these preliminary checks, the dataset was considered suitable for CFA using Maximum Likelihood estimation.

### Data analysis

To evaluate the reliability and validity of the SCLC questionnaire, the data were analyzed in two steps. Only the fully completed questionnaires were included in the analysis. Initially, the internal consistency of the questionnaire was assessed, followed by a confirmatory factor analysis (CFA) to evaluate its construct validity. We used similar statistical procedures as earlier validation studies [17].

### Confirmatory factor analysis (CFA)

CFA was conducted to evaluate the construct validity of the questionnaire using the generalized least squares method. To assess overall model fit, Chi-square ($\chi^2/df$), Comparative Fit Index (CFI), Goodness of Fit Index (GFI), Adjusted Goodness of Fit Index (AGFI), and Root Mean Square Error of Approximation (RMSEA) were utilized. Acceptable fit criteria included $\chi^2/df$ values between 1 and 3, CFI > 0.97, GFI > 0.95, AGFI > 0.8, SRMR < 0.08 and RMSEA < 0.08. Windows IBM SPSS Amos 29 Graphics was used for the analysis [24,28,29].

### Model fit

Prior to analysis, we established criteria for considering item deletion based on both statistical and conceptual grounds. Items with consistently very low inter-item correlations ($r < 0.30$) with the other items were flagged as potentially problematic, as such low correlations suggest that the item may not adequately capture the intended construct and could reduce internal consistency [28]. In addition, content validity was assessed by examining whether the item conceptually fit within the defined dimensions of satisfaction and self-confidence. Only items that demonstrated both weak statistical performance (e.g., inter-item correlations < 0.30) and poor conceptual alignment were considered for removal. Based on these criteria, ITEM 13 was removed due to very low and even negative correlations with other items, while ITEM 10 was removed due to its marginal correlations combined with a conceptual mismatch with the intended constructs.

Subsequently, we attempted to improve model fit by inspecting the modification indices (MI) and adding error covariances between items within the same factor where theoretically justified. Specifically, error terms were allowed to correlate for item pairs with MI values between 5–6 and subsequently between 6–7, provided the items belonged to the same latent construct. Based on these criteria, correlated error terms were added between items 4–5, 1–2, and 2–5 respectively within the Satisfaction factor [28].

Furthermore, Composite Reliability (CR) and Average Variance Extracted (AVE) were calculated for the adjusted model. Internal consistency reliability was assessed using CR rather than Cronbach's alpha, because the assumptions of tau-equivalence and uncorrelated errors required for alpha were not met, as shown in the CFA output. CR accounts for varying factor loadings and correlated error terms, providing a more accurate estimate of reliability in the context of confirmatory factor analysis. CR values ≥ 0.70 were considered acceptable. In addition, AVE was calculated to assess convergent validity of the constructs, with values ≥ 0.50 indicating that, on average, the construct explains at least half of the variance of its indicators [28,30,31].

## Results

### Demographic characteristics

The demographic and employment characteristics of the pharmacy technicians are shown in Table 2. Of the 230 pharmacy technicians invited, 129 completed the questionnaire in full, and participants who did not complete the questionnaire

**Table 2. Demographic and employment characteristics of pharmacy technicians.**

| Characteristic | Frequency | Percent |
|---|---|---|
| Participants (N) =129 | | |
| **Gender** | | |
| Female | 118 | 91.5% |
| Male | 3 | 2.3% |
| No answer | 8 | 6.2% |
| **Age** | | |
| Under 20 years | 1 | 0.8% |
| 20–30 years | 34 | 26.4% |
| 30–40 years | 38 | 29.5% |
| 40–50 years | 29 | 22.5% |
| Over 50 years | 27 | 20.9% |
| **Years as a Pharmacy Technician** | | |
| 0–5 years | 33 | 25.6% |
| 5–10 years | 20 | 15.5% |
| 10–20 years | 30 | 23.3% |
| More than 20 years | 46 | 35.7% |
| **Years at Erasmus MC as a Pharmacy Technician** | | |
| 0–5 years | 66 | 51.2% |
| 5–10 years | 34 | 26.4% |
| 10–20 years | 12 | 9.3% |
| More than 20 years | 17 | 13.2% |
| **Employment Status** | | |
| Part-time (8–32 hours) | 102 | 79.1% |
| Full-time (36 hours) | 27 | 20.9% |

were excluded from the analysis. Among the participants, 93% were female. All age groups were represented, and 79% worked part-time. Approximately half of the respondents had worked at Erasmus MC for less than five years, while 36% had more than 20 years of experience overall as pharmacy technicians.

### Psychometric analysis of the satisfaction and self-confidence in learning scale

**Reliability.** For the assessment of reliability, both Composite Reliability (CR) and Average Variance Extracted (AVE) were calculated. Table 3 presents the CR and AVE values (with recommended reference values) for each construct, based on the adjusted model in which Items 10 and 13 were removed and error terms within the Satisfaction construct were correlated based on modification indices (MI) between 5 and 6 and then between 6–7. These results reflect the final adjusted model. The CR and AVE values for the unadjusted model with all 13 items, as well as for the adjusted model with Items 10 and 13 removed but without correlating error terms, are provided in the Supporting Information for comparison.

While the AVE for Satisfaction was slightly below the recommended cutoff of 0.50, it approached adequacy and, combined with the acceptable CR, suggests that the items adequately reflect the underlying constructs [28].

**Construct validity.** The two latent constructs, Satisfaction and Self-confidence, demonstrated a weak positive correlation (r = 0.24), suggesting that they are conceptually related yet sufficiently distinct to support discriminant validity (Fig 1).

**Table 3. Composite reliability (CR) and average variance extracted (AVE) for the satisfaction and self-confidence factors (including reference values).**

| Factor | Composite Reliability | Average Variance Extracted |
|---|---|---|
| Satisfaction | 0.80 (<0.8) | 0.45 (>0.5) |
| Self-confidence | 0.85 (<0.8) | 0.50 (>0.5) |

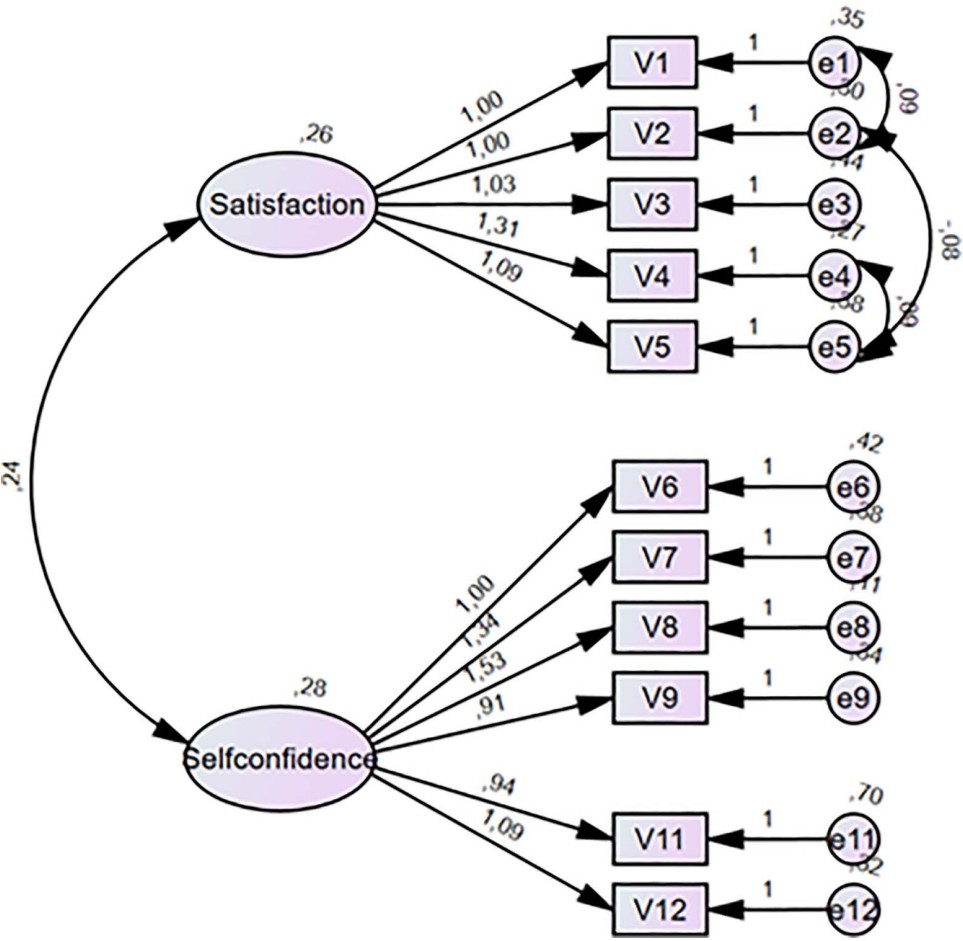

**Fig 1. Path diagram of the adjusted SCLC model.** Items 10 and 13 are removed and MI-based error term correlations are made (> 6, then > 5) within the construct satisfaction.

The fit indices of the confirmatory factor analysis (CFA) indicated a moderate fit to the data: CFI = 0.92, GFI = 0.87, AGFI = 0.79, RMSEA = 0.109, RMR = 0.045, and $\chi^2$/df = 2.51 ($\chi^2$ = 97.83, df = 39, p < 0.001). While the CFI, RMR, and $\chi^2$/df met conventional thresholds for acceptable fit, the GFI, AGFI, and RMSEA suggested room for improvement in model fit. Detailed fit indices and reference thresholds are presented in Table 4 [28,32].

The Supporting Information, S1 Table presents the inter-item correlations of the full model (13 items), while S2-S5 Tables provide the fit indices, as well as the CR and AVE, for both the full model (13 items) and the adjusted 11-item model, without MI-based modifications. The path diagrams of both models are shown in S6 and S7 Figs.

**Table 4. The fit indices for the confirmatory model.**

| Index | Value | Values for good model fit |
|---|---|---|
| CFI | 0.92 | >0.97 (>0.9 is traditional) |
| GFI | 0.87 | >0.95 |
| AGFI | 0.79 | >0.80 |
| RMSEA | 0.109 | <0.08 good; 0.5−.10 moderate |
| RMR | 0.045 | <0.05 |
| Goodness of fit test | $\chi^2 = 97,831$; df = 39; p < 0.000 | |
| Reason for fit | $\chi^2/ gl = 2.51$ | |

This table presents the indices of goodness of fit for the model, alongside reference values for a good model fit in the right-hand column. The indices include the Comparative Fit Index (CFI), Goodness of Fit Index (GFI), Adjusted Goodness of Fit Index (AGFI), Root Mean Square Error of Approximation (RMSEA), and Root Mean Square Residual (RMR), with values compared to standard thresholds. The chi-square test for goodness of fit and the chi-square to degrees of freedom ratio ($\chi^2$/df) are also reported. Fit-indices are calculated where ITEM 10 and ITEM 13 are removed and MI-based error term correlations (> 6, then >5) are made within the construct Satisfaction.

## Discussion

### Reliability and construct validity

The findings of our study indicate that the adaptation and validation of the SCLC for Dutch pharmacy technicians yielded acceptable psychometric properties, supporting its use in this context.

The reliability and validity findings of the Dutch version of the SCLC are largely consistent with previous validation studies conducted in other languages and settings. In our study, the adjusted model showed acceptable internal consistency, with Composite Reliability (CR) values of 0.80 for the Satisfaction factor and 0.85 for the Self-confidence factor. Average Variance Extracted (AVE) was 0.45 for Satisfaction and 0.50 for Self-confidence, indicating marginally sufficient convergent validity for Satisfaction and adequate for Self-confidence. While our AVE for Satisfaction fell just below the commonly suggested threshold of 0.50, similar patterns have been observed in previous research, particularly for constructs measured by relatively few items with moderate loadings [28].

Direct comparisons with earlier studies are somewhat difficult, as most prior validation studies reported Cronbach's alpha rather than CR. Cronbach's alpha assumes tau-equivalence and uncorrelated errors, whereas CR, as used here, does not and is therefore more appropriate when these assumptions are not met. Nevertheless, the CR values we found are comparable to the Cronbach's alpha values reported in other language adaptations, which ranged from 0.77 to 0.94 for the Satisfaction and Self-confidence factors [11,14–18,33].

The model fit indices in our study indicated an overall moderate fit to the data. The CFI (0.92) and $\chi^2$/df (2.51) fell within acceptable ranges (Schreiber et al., 2006), while the RMSEA (0.109) and GFI (.87) were less optimal. This pattern of results is comparable to findings in other SCLC validation studies that also reported acceptable CFI values (≥0.90) alongside less favorable RMSEA and GFI values, reflecting the challenges in achieving excellent fit in scales measuring subjective constructs in diverse educational contexts [16–18]. Taken together, the reliability and model fit of the Dutch SCLC are in line with previous validation studies and support the use of the scale in the Dutch context.

In contrast to studies by Almeida et al. and Tosterud et al. [15,18], which validated the SCLC in the context of a single educational simulation, our study encompassed a broader range of learning environments, including both digital modules and clinical lessons. This more diverse context may have influenced participants' interpretation of certain items, particularly those that refer to instructor behavior or learning responsibility. For instance, ITEM 13 ('It is the responsibility of the instructor/teacher to tell me what I need to learn from the learning activities') may be understood differently in a digital learning environment, where the 'instructor' is not a person but embedded within the system (e.g., video content or

interactive modules). Similarly, ITEM 9 ('The instructors/teachers used useful resources to teach me things') may carry different connotations depending on whether the participant is reflecting on a live clinical lesson or a self-paced e-learning environment.

These contextual differences could have affected the factor loadings and overall fit indices of the model. This reinforces the idea that attitudes toward learning—and by extension, constructs such as satisfaction and self-confidence—are not only theoretically but also contextually dependent. This underlines the relevance of adapting instruments like the SCLC not only linguistically, but also with regard to their application context, especially when transitioning from simulation-based to mixed or digital learning environments.

## Practical implications

A validated instrument to assess pharmacy technicians' satisfaction and self-confidence in learning is essential to understanding how educational interventions contribute to professional development. The SCLC, originally designed for nursing education, has been adapted in this study to fit the linguistic, professional, and contextual needs of Dutch pharmacy technicians. Our adaptation acknowledges the shift in educational formats within healthcare—from traditional simulations and in-person lessons to increasingly digital and blended learning environments.

The successful bifactorial structure, revealed through Confirmatory Factor Analysis, supports the continued use of the two original subscales—satisfaction with current learning and self-confidence in learning—as either separate measures or as a single composite score. The moderate correlation between the two factors ($r = 0.24$) is consistent with the conceptual understanding that, while related, satisfaction and self-confidence represent distinct but complementary aspects of the learning experience [11,17].

This distinction has important implications for the design and evaluation of professional learning activities. For example, an educational program may be perceived as satisfying in terms of structure and relevance but may not significantly boost learners' confidence in applying knowledge in practice. Conversely, activities that challenge learners may increase confidence but yield lower satisfaction if they are perceived as too demanding. By measuring both constructs independently, educators and trainers can more accurately identify areas for improvement.

Notably, ITEM 10 ('It is my responsibility as a pharmacy technician to learn what is required from the learning activities') and ITEM 13 ('It is the responsibility of the instructor/teacher to tell me what I need to learn from the learning activities') performed inconsistently. These items relate to the learner's perceived role and autonomy in the educational process. While prior validations among students showed adequate item performance, our findings among employed pharmacy technicians suggest that these respondents may interpret the concept of learning responsibility differently. In vocational settings, pharmacy technicians may not always view further education as a personal obligation, particularly when it is not structurally embedded in their professional routines. This reflects broader challenges in promoting lifelong learning within the profession, as also noted in earlier literature [1–7].

Furthermore, the interpretation of "instructor" in digital learning environments is less straightforward than in traditional education. Participants may associate this term with written guidelines, videos, or even automated feedback, rather than with a live teacher. This ambiguity may have contributed to the poor item fit. These findings highlight how learner perceptions of autonomy and instruction can vary across contexts and professions, which should be considered when applying instruments like the SCLC in non-student, workplace-based populations.

Given these observations, we recommend removing ITEM 10 and ITEM 13 and use an adapted 11-item version of the SCLC when applied in digital or mixed learning formats, particularly among employed healthcare professionals.

## Strengths and limitations

A key strength of this study lies in its sampling strategy. By targeting employed pharmacy technicians in the Netherlands, we included a population that closely reflects the real-world demographic and professional characteristics of

the workforce, particularly in terms of gender and age distribution [30]. This is important because attitudes toward learning—especially in the context of workplace-based continuing education—are influenced not only by educational design, but also by factors such as age, years of experience, and perceived relevance to professional practice [7,32].

Unlike many prior studies using the SCLC, which focused on nursing students in training, our study involved practicing healthcare professionals for whom participation in structured learning is often less habitual or even optional. This adds value to our findings, as it highlights how learning satisfaction and self-confidence manifest among healthcare workers with varied professional responsibilities and time constraints. Assessing these constructs in a working population contributes to the broader goal of fostering lifelong learning and improving job satisfaction within healthcare teams—objectives that are increasingly vital in today's dynamic care environment [1–3,20,21].

However, our study also has limitations. Most notably, the sample size of 129 participants, while in line with the general guideline of 10 respondents per item [14], falls short of more conservative thresholds for confirmatory factor analysis. Previous methodological studies recommend a minimum of 200 participants for robust CFA results, especially when multiple interdependent parameters are involved [29,34,35]. While our model fit indices were moderate overall, it is possible that a larger sample would yield more stable or differentiated factor solutions. Furthermore, although RMSEA values tend to be stable across sample sizes, they can be overly sensitive in smaller samples, potentially rejecting models that are otherwise theoretically sound. This may explain why the RMSEA value in the present study was not ideal, despite the model showing good theoretical and empirical fit in other respects [36].

## Future research

While the adapted Dutch version of the SCLC demonstrates acceptable reliability and moderate validity, future research is warranted to further optimize and substantiate the instrument's psychometric properties. First, alternative model specifications could be explored to improve model fit. Possible directions include testing a bifactor model, where a general factor alongside specific factors for satisfaction and self-confidence is estimated; a higher-order factor model, with an overarching latent construct underlying the two dimensions; or a three-factor model, reintroducing responsibility-related items as a separate dimension. An exploratory structural equation modeling (ESEM) approach could also be considered, as it allows for cross-loadings and may provide a more realistic representation of the construct structure.

Second, larger sample sizes would enable more robust analyses and help assess the stability of the model fit indices, particularly the RMSEA, which is known to be sensitive to sample size [23]. Future studies with larger and more diverse samples could provide more precise estimates and potentially reveal better fit.

Third, conducting the study in different educational or professional settings could help examine the generalizability of the instrument and whether key terms, such as "instructor" and "teacher," are interpreted consistently across contexts, including fully online learning environments.

Taken together, these future directions could provide deeper insight into the dimensionality, validity, and applicability of the Dutch SCLC and support its ongoing refinement.

## Conclusion

This study suggests that the Dutch version of the SCLC is a moderately reliable and valid tool for assessing pharmacy technicians' satisfaction with education and self-confidence in learning. Given the absence of other validated instruments in this context, this scale offers a useful starting point for evaluating and improving educational programs. However, as the psychometric properties indicated room for improvement and the model fit was moderate, further research with larger samples is needed to refine and confirm the suitability of this questionnaire.

## Supporting information

**S1 Table. Inter item correlations of the 13- item model without adjustments.** This table presents Pearson correlation coefficients between individual items of the SCLC questionnaire. Significant correlations are denoted by $p < 0.05$ (*), $p < 0.01$ (**). High correlations can be observed between items such as ITEM 7 and ITEM 8 ($r = 0.724$), and ITEM 4 and ITEM 5 ($r = 0.658$**), suggesting strong relationships within certain subscales. ITEM 13R, a reverse-coded item, generally shows weaker or negative correlations with other items. This table illustrates the internal consistency and inter-item relationships across the questionnaire.
(DOCX)

**S2 Table. Fit-indices of the full 13-item model without adjustments.** The indices of goodness of fit for the model, alongside reference values for a good model fit in the right-hand column. The indices include the Comparative Fit Index (CFI), Goodness of Fit Index (GFI), Adjusted Goodness of Fit Index (AGFI), Root Mean Square Error of Approximation (RMSEA), and Root Mean Square Residual (RMR), with values compared to standard thresholds. The chi-square test for goodness of fit and the chi-square to degrees of freedom ratio ($\chi^2$/df) are also reported.
(DOCX)

**S3 Table The composite reliability (CR) and average variance extracted (AVE).** CR and AVE are calculated for the full 13-item model without adjustments.
(DOCX)

**S4 Table Fit-indices of the adjusted 11 item model.** This table presents the indices of goodness of fit for the model, alongside reference values for a good model fit in the right-hand column. The indices include the Comparative Fit Index (CFI), Goodness of Fit Index (GFI), Adjusted Goodness of Fit Index (AGFI), Root Mean Square Error of Approximation (RMSEA), and Root Mean Square Residual (RMR), with values compared to standard thresholds. The chi-square test for goodness of fit and the chi-square to degrees of freedom ratio ($\chi^2$/df) are also reported. Fit-indices are calculated where ITEM 10 and ITEM 13 are removed- without MI-based modifications and without correlated error terms.
(DOCX)

**S5 Table The Composite Reliability (CR) and Average Variance Extracted (AVE).** CR and AVE are calculated for the adjusted 11- item model – where ITEM 10 and ITEM 13 are removed- without MI-based modifications and without correlated error terms.
(DOCX)

**S6 Fig. Path diagram of the SCLC model (entire scale, no items removed).** No MI-based error correlations are made.
(TIF)

**S7 Fig. Path diagram of the adjusted SCLC model (11-itemscale, ITEM 10 and ITEM 13 are removed).** No MI- based error correlations are made.
(TIF)

**S8 File. Original SCLC questionnaire. Available from the National League for Nursing. The questionnaire is owned and copyrighted by the National League for Nursing.**
(PDF)

**S9 File. Translated and adapted Dutch version of the SCLC questionnaire.**
(PDF)

**S10 File. Raw data of this study (Excel file).**
(XLSX)

## Acknowledgments

We thank all participants at both pharmacies for their contribution to this study.

## Author contributions

**Conceptualization:** Narin Akrawi, Wim J.R. Rietdijk, Floor van Rosse, Heleen van der Sijs.

**Data curation:** Narin Akrawi.

**Formal analysis:** Narin Akrawi, Wim J.R. Rietdijk.

**Investigation:** Narin Akrawi.

**Methodology:** Narin Akrawi, Wim J.R. Rietdijk, Floor van Rosse, Heleen van der Sijs.

**Supervision:** Birgit C.P. Koch, Floor van Rosse, Heleen van der Sijs.

**Validation:** Narin Akrawi.

**Writing – original draft:** Narin Akrawi.

**Writing – review & editing:** Narin Akrawi, Wim J.R. Rietdijk, Birgit C.P. Koch, Floor van Rosse, Heleen van der Sijs.

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
