## [Decision Letter · Decision Letter 0]

23 Jun 2025

Dear Dr. Akrawi,

Thank you for submitting your manuscript to PLOS ONE. After careful consideration, we feel that it has merit but does not fully meet PLOS ONE’s publication criteria as it currently stands. Therefore, we invite you to submit a revised version of the manuscript that addresses the points raised during the review process.

**ACADEMIC EDITOR:**

Concerns:

The manuscript fails to clearly indicate where Table 1, Table 2, Table 3, Table 4, and Figure 1 should be inserted in the main text.The RMSEA value of 0.099 exceeds the commonly accepted threshold for good model fit (<0.08). This raises concerns about the robustness of the confirmatory factor analysis model, especially given the small sample size (N=129).The authors should explain in advance the criteria used for considering item deletion.The terms like "instructor" and "teacher" may be interpreted differently in digital learning. This ambiguity could be mitigated by clarifying in the methods or providing alternative wording tested during pilot phases.

We look forward to receiving your revised manuscript.

Kind regards,

Mohd Ismail Ibrahim, MCom.Med

Academic Editor

PLOS ONE

Journal Requirements:

3. Please include captions for your Supporting Information files at the end of your manuscript, and update any in-text citations to match accordingly. Please see our Supporting Information guidelines for more information: http://journals.plos.org/plosone/s/supporting-information .

Reviewers' comments:

Reviewer's Responses to Questions

**Comments to the Author**

1. Is the manuscript technically sound, and do the data support the conclusions?

Reviewer #1: Partly

Reviewer #2: Yes

2. Has the statistical analysis been performed appropriately and rigorously?

Reviewer #1: Yes

Reviewer #2: No

3. Have the authors made all data underlying the findings in their manuscript fully available?

Reviewer #1: No

Reviewer #2: Yes

4. Is the manuscript presented in an intelligible fashion and written in standard English?

Reviewer #1: No

Reviewer #2: Yes

Reviewer #1: the manuscript is titles The Validity and Reliability of the Dutch Version of the Student Satisfaction and Self Confidence in Learning Scale (SCLC) for Pharmacy Technicians.

THIS IS NOT noval work AND authors used old data. references must be added of 2025. there are no figures and diagrams presented in the manuscript.

Reviewer #2: The paper is aimed at assessing the validity and reliability of Dutch adapted SCLC. The paper is well written. Nevertheless, some methodological aspects have drawn my concern.

1. The authors used Cronbach’s alpha (CA) for assessing reliability and CFA-loading factor for validity. CA assumes a tau-equivalent measurement model in which factor loadings are equal and the measurement errors are not correlated. These contrast with the CFA results. Thus, the CA use is invalid. I suggest that the authors further use the CFA results for deriving composite reliability (CR) or construct reliability index. The authors may refer to Hair et al. (2019).

2. The authors calculated Cronbach’s alpha of total questionnaire. This is meaningless, since a reliability index is associated only to one latent construct, dimension, or subdimension.

3. The authors treated the Likert scale data as continuous normal data. Likert scale data has a limited number of distinct values. Naturally, it is ordinal data. And by the equidistance assumption, it can be upgraded into discrete interval data which is not a normal data. To improve the validity of the analysis, I suggest the author to use polychoric correlation matrix supplemented by its asymptotic covariance matrix as the inputs and robust maximum likelihood as the estimation method and corrected chi-square statistics for non-normality in the CFA analysis.

4. The reported main fit indices indicate an insufficient model fit. I suggest the authors to accommodate modification indices higher than 5 or 6 instead of 20 step-wisely. Report the CFA results after the estimates provide a good model fit. Please refer for Hair et al. (2019) for the criteria of a good model fit.

Hair, J. F., Black, W. C., Babin, B. J., & Anderson, R. E. (2019). Multivariate Data Analysis (8th ed.). Cengage.

**Do you want your identity to be public for this peer review?** For information about this choice, including consent withdrawal, please see our Privacy Policy

Reviewer #1: No

Reviewer #2: **Yes: ** Yusep Suparman

---

## [Author Response · Author response to Decision Letter 1]

22 Jul 2025

Dear reviewers and editor,

Thank you very much for your constructive feedback and valuable suggestions, which have helped us to improve the quality and clarity of our manuscript.

Please find in the file named 'Response to reviewers' our point-by-point responses to each of the reviewers' and editor’s comments, along with a summary of the corresponding revisions made to the manuscript.

Should you have any questions or may you need any clarification, please let us know.

Kind regards,

Narin Akrawi

---

## [Decision Letter · Decision Letter 1]

12 Aug 2025

The Validity and Reliability of the Dutch Version of the Student Satisfaction and Self-Confidence in Learning Scale (SCLC) for Pharmacy Technicians

PONE-D-25-23816R1

Dear Dr. Akrawi,

We’re pleased to inform you that your manuscript has been judged scientifically suitable for publication and will be formally accepted for publication once it meets all outstanding technical requirements.

Kind regards,

Mohd Ismail Ibrahim, MCom.Med

Academic Editor

PLOS ONE

Additional Editor Comments (optional):

The revised manuscript demonstrates that the authors have thoroughly addressed all reviewer and editorial comments, resulting in notable improvements to the clarity, structure, and methodological rigor of the study. The peer reviewers concur with the decision to accept, acknowledging the work’s methodological soundness and contribution to the field.

Reviewers' comments:

Reviewer's Responses to Questions

**Comments to the Author**

Reviewer #1: All comments have been addressed

2. Is the manuscript technically sound, and do the data support the conclusions?

Reviewer #1: Partly

3. Has the statistical analysis been performed appropriately and rigorously?

Reviewer #1: Yes

4. Have the authors made all data underlying the findings in their manuscript fully available?

Reviewer #1: Yes

5. Is the manuscript presented in an intelligible fashion and written in standard English?

Reviewer #1: Yes

Reviewer #1: This study suggests that the Dutch version of the SCLC is a moderately reliable and valid tool

for assessing pharmacy technicians’ satisfaction with education and self-confidence in learning. Given

the absence of other validated instruments in this context, this scale offers a useful starting point for

evaluating and improving educational programs. However, as the psychometric properties indicated

room for improvement and the model fit was moderate,.Further research with larger samples is needed to refine and confirm the suitability of this questionnaire.This conclusion seems good for acceptance.

**Do you want your identity to be public for this peer review?** For information about this choice, including consent withdrawal, please see our Privacy Policy

Reviewer #1: No

---

## [Editor Report · Acceptance letter]

PONE-D-25-23816R1

PLOS ONE

Dear Dr. Akrawi,

I'm pleased to inform you that your manuscript has been deemed suitable for publication in PLOS ONE. Congratulations! Your manuscript is now being handed over to our production team.

Kind regards,

on behalf of

Dr. Mohd Ismail Ibrahim

Academic Editor

PLOS ONE